# Inhibin B-to-Anti-Mullerian Hormone Ratio as Noninvasive Predictors of Positive Sperm Retrieval in Idiopathic Non-Obstructive Azoospermia

**DOI:** 10.3390/jcm12020500

**Published:** 2023-01-07

**Authors:** Chenyao Deng, Defeng Liu, Lianming Zhao, Haocheng Lin, Jiaming Mao, Zhe Zhang, Yuzhuo Yang, Haitao Zhang, Huiyu Xu, Kai Hong, Hui Jiang

**Affiliations:** 1Department of Urology, Peking University Third Hospital, Beijing 100191, China; 2Department of Andrology, Peking University Third Hospital, Beijing 100191, China; 3Department of Reproductive Medicine Center, Peking University Third Hospital, Beijing 100191, China; 4Department of Human Sperm Bank, Peking University Third Hospital, Beijing 100191, China

**Keywords:** idiopathic non-obstructive azoospermia, reproductive hormones, sperm retrieval, inhibin B, Anti-Mullerian hormone

## Abstract

Background: The lack of clinically useful biomarkers for predicting micro-TESE outcomes in males with idiopathic NOA. To find clinically reliable serum reproductive hormone markers to predict the outcome of sperm retrieval in men with idiopathic NOA undergoing micro-TESE. Methods: We analyzed the clinical data of 168 men with idiopathic NOA treated with micro-TESE. The clinical predictors of a successful sperm retrieval were put to the test using logistic regression analysis. The clinical net benefit was calculated using a decision-curve analysis, and the predictive power of each variable was assessed using the ROC-derived area under the curve. Result: Between positive group and negative group for sperm retrieval, there was a statistically significant difference in INHB, AMH, and INHB/AMH. AMH, INHB, and INHB/AMH were each independent predictors of successful sperm retrieval, with INHB (OR1.02, *p* = 0.03), AMH (OR0.85, *p* = 0.01), INHB/AMH (OR1.08, *p* < 0.01). The ROC curve determined the optimal cut-off values for serum INHB and INHB/AMH in positive sperm retrieval patients undergoing micro-TESE. 21.51 pg/mL was the cut-off value for INHB. The cut-off value for INHB/AMH was 3.19, which had a 86.3% sensitivity and a 53.8% specificity. Using INHB and INHB/AMH prior to micro-TESE sperm retrieval in idiopathic NOA patients improved the net benefit of positive sperm retrieval, and the net benefit score of INHB/AMH was better than that of INHB, according to decision analysis curves. Conclusion: Serum INHB have predictive value for sperm retrieval outcomes in idiopathic NOA patients undergoing micro-TESE. Combining INHB and AMH, INHB/AMH seems to be a better predictor.

## 1. Introduction

Male infertility is described as a male-related female’s inability to naturally conceive within a year after having regular sexual contact and without utilizing contraception. Azoospermia, one of the most serious forms of male infertility, is the complete absence of sperm in the ejaculated semen and accounts for about 5% to 20% of male infertility patients [1]. Azoospermia is categorized into obstructive and non-obstructive. Obstructive azoospermia mostly causes the absence of sperm in the ejaculate due to post-testicular ducts occlusion. Non-obstructive azoospermia (NOA) is mostly associated with impaired spermatogenesis and accounts for >60% of azoospermia1. The most severe case of infertility in males is NOA. Part of them can only produce genetic offspring by surgically obtaining mature sperm from the testes.

The hypothalamic-pituitary-gonadal (HPG) axis is important in regulating sperm production and maturation. Hypothalamic gonadotropin-releasing hormone (GnRH) stimulates gonadotropin secretion, namely follicle stimulating hormone releasing hormone (FSH) and luteinizing hormone (LH), from the anterior pituitary, which in turn promotes the secretion of Inhibin B (INHB), Anti-Mullerian Hormone (AMH), and testosterone (T) from the testes. Therefore, these gonadotropins are the main players in spermatogenesis development and maintenance (Figure 1). INHB is a dimeric glycoprotein hormone that is released by testicular support cells and plays an important role in the negative feedback control of the HPG axis in males. Serum INHB levels were negatively correlated with FSH in adult males, acting as negative feedback on FSH. INHB and AMH are secreted specifically by testicular support cells and may be more useful in predicting normal spermatogenesis in the testis than FSH. A prospective study of 139 males revealed that AMH and INHB have prognostic value for sperm extraction in patients with NOA; however, Emma et al. reported contradicting results [2]. Serum INHB and AMH as sensitive predictors of effective testicular sperm retrieval in NOA remain debatable although research has indicated that serum INHB and AMH are more sensitive predictors of testicular spermatogenesis than serum FSH [3].

Microdissection testicular sperm extraction (Micro-TESE) is the gold standard treatment for males with idiopathic NOA [4]. In idiopathic NOA, the spermatogenesis of the testis is severely unbalanced, and there are even differences in the spermatogenic function of different segments of the seminiferous tubules in the same lifetime. This increases the difficulty of obtaining spermatozoa to a higher degree. The currently reported sperm retrieval rate (SRR) for idiopathic NOA is approximately 30% [5]. Identifying the spermatozoa in the vas deferens throughout the procedure is crucial for the success of micro-TESE. Micro-TESE has several benefits over conventional testicular puncture or incision for sperm extraction. First, micro-TESE provides greater intraoperative testicular tissue exposure, which facilitates the discovery of localized spermatogenic areas and significantly improve SRR. Second, the use of a microscope can minimize postoperative bleeding and difficulties associated with testicular atrophy while preventing harm to the testicular blood supply system to the maximum extent possible. However, post-micro-TESE complications, including postoperative scrotal hematoma, scrotal swelling and discomfort, and infection, cannot be ignored due to the greater testicular tissue exposure. The study discovered that micro-TESE reduced blood T levels while increasing serum LH and FSH levels, with T recovering to baseline levels after 12 months, while postoperative T levels in patients with kernicterus only recovered to 50% of preoperative levels [6]. Negative sperm retrieval results can also impact the male’s self-esteem, erectile function, intercourse satisfaction, orgasmic function, and relationship with their couple [7]. Numerous investigations have demonstrated that the success of micro-TESE cannot yet be predicted by current reproductive hormones, age, testicular volume, or testicular disease diagnosis [3,8,9]. The predictive usefulness of the relevant preoperative markers remains equivocal [10].

Given the scarcity of clinically useful biomarkers for predicting micro-TESE outcomes in males with idiopathic NOA, we aimed to explore the predictive value of serum INHB and AMH in predicting sperm recovery in males with idiopathic NOA receiving micro-TESE. This study investigated the differences in preoperative reproductive hormones between the micro-TESE positive and negative groups, as well as the prediction ability of INHB and INHB/AMH for successful micro-TESE sperm retrieval.

## 2. Materials and Methods

### 2.1. Study Design

This single-center prospective study was conducted from January 2019 to December 2020 and followed up on the clinical results of micro-TESE in male patients with idiopathic NOA who were seen in our Center for Reproductive Medicine’s outpatient clinic. We gathered and counted clinical information on age, testicular volume, and reproductive hormones as well as the clinical outcomes of sperm discovered by micro-TESE. Semen will be taken once more on the day of the micro-TESE retrieval to confirm azoospermia in all patients who performed at least three preoperative centrifugal semen tests that demonstrated azoospermia. Idiopathic NOA was defined after comprehensive diagnostic evaluations of all know causes for non-obstructive azoospermia. Patients who met the following criteria were excluded: (1) chromosomal disorders previously linked to azoospermia, such as AZFa/b/c microdeletions on the Y chromosome, Klinefelter syndrome, or Kallman syndromes; mutations of the cystic fibrosis conductance regulator gene linked to congenital bilateral absence of the vas deferens; (2) hypothalamic/pituitary defects; (3) testicular tumors; (4) testicular factors (cryptorchidism, varicocele, and disturbances of erection/ejaculation) linked to infertility; (5) using drugs that alter hormone levels (e.g., exogenous testosterone, selective estrogen receptor modulators, gonadotropins, or aromatase inhibitors); (6) either testicular or pituitary surgery, or a previous vasectomy.

### 2.2. Demographic Data

We gathered demographic and clinical information on the patients, including age, medical history, testicular fixation history, and radiotherapy or chemotherapy history. Clinical indicators included blood hormone levels, testicular volume (measured by a testicular meter on the pierced side of the testes), and semen analysis results (AMH, INHB, FSH, LH, prolactin (PRL), T, and 17β-estradiol [E2]). INHB/AMH is defined as the ratio obtained by dividing the absolute value of INHB by the absolute value of AMH.

### 2.3. Hormonal Profile

Each patient underwent venous blood sampling from 7 a.m. to 11 a.m. following an overnight fast. AMH (reference range = 2.04–19.22 ng/mL), PRL (reference range = 2.04–19.22 ng/mL), T (reference range = 3.4–5.2 ng/mL), FSH (reference range = 0.96–11.55 mIU/mL), LH (reference range = 0.63–11.70 mIU/mL), INHB (reference range = 15–295 pg/mL), and E2 (reference range = 18–72 pg/mL) were evaluated using chemiluminescent immunoassay. All patients went to the same lab, and Guangzhou Kangrun Biotechnology Company furnished a uniform batch of the chemicals utilized in the assay.

### 2.4. Micro-TESE Procedure

The micro-TESE method previously published by Schlegel et al. [11]. A longitudinal incision is made in the middle of the scrotum; the skin, meatus, and sheath are incised; and the testes and epididymis are exposed and extruded. Under an operational microscope (S88, Carl Zeiss, Germany) with a maximum magnification of 18×, the testicular parenchyma was meticulously dissected layer by layer while the albuginea was carefully cut along the middle transverse surface of the testis. The relatively full and thick spermatogenic tubules were chosen, put in a Petri plate, and delivered right away to the embryo laboratory. Two senior laboratory staff members perform the procedure by slicing, tearing, and separating testicular tissue with a 1-mL syringe needle to release spermatogenic cells and sperm from the germinal tubules and create a cell solution that is examined under a microscope for mature sperm. The procedure is finished if adequate sperm are found in a sufficient amount and with good morphology. The next testicle is incised and carefully examined simultaneously if one testicle is empty of sperm. Interrupted 5-0 silk sutures are used to seal the incision in the white membrane. The surgical incision was closed, the testis was moved back into the sheath, and the scrotum was tightly bandaged. The obtained germinal tubules were maintained in culture for 24 h before conducting another sperm search when no spermatozoa were discovered intraoperatively in a patient. The germinal tubules were then cultured if no sperm were still discovered. In the absence of sperm, the culture fluid was centrifuged and the search for sperm was repeated.

### 2.5. Statistics

The continuous variables are expressed as the median and interquartile range (IQR). Continuous variables in two groups were compared using the Mann–Whitney test, with two-by-two comparisons using the Bonferroni method to adjust for significance levels. The chi-square test was used for the comparison of rates. Statistical analysis consists of several steps. The odds ratio (OR) and 95% confidence interval (CI) of the univariate and multivariate associations between age, testicular volume, FSH, LH, T, PRL, E2, AMH, INHB, INHB/AMH, and micro-TESE positive sperm retrieval were first assessed using binary logistic regression models after adjusting for prior covariates. Receiver operating characteristic curves were used to quantify the predictive accuracy (area under the curve [AUC]) of INHB and INHB/AMH, and the Jorden index to determine the cutoff point. Finally, we used decision curve analysis (DCA) to assess the net clinical benefit of predictive markers. Statistical analysis was performed using GraphPad Prism 5.9(Graphpad software, USA). DCA was calculated using R version 3.3.0. All tests were two-way, and *p*-values of <0.05 were considered statistically different.

### 2.6. Ethical Approval

Data collection follows the principles outlined in the Declaration of Helsinki. All patients agreed to donate their anonymous information and tissue samples for upcoming investigations by signing an informed consent form. The study was approved by the Ethics Review Committee of Reproductive Medicine of Peking University Third Hospital, China (No. 2017SZ-035). Written informed consent was obtained from all patients before the commencement of the study. All patient information was anonymized and de-identified before analysis.

## 3. Results

Table 1 shows the descriptive information for the complete patient group. The classification was made based on the outcomes of surgical sperm capture using micro-TESE. There were 51 sperm retrievals that were positive (SRR = 30.4%). Statistically significant differences were found between the two groups in INHB, AMH, INHB/AMH, and testicular volume, and without significant differences between age, FSH, LH, T, PRL, and E2. AMH was 3.87 (2.05–7.73) ng/mL in the micro-TESE negative group, which was significantly higher than the positive group at 2.36 (0.65–4.80) ng/mL (*p* = 0.007). INHB in the micro-TESE negative group was 9.98 (5.00–16.27) pg/mL, which is significantly lower than that in the positive group at 17.89 (8.00–36.00) pg/mL (*p* = 0.001). INHB/AMH was significantly lower in the micro-TESE negative group than in the positive group (*p* < 0.001).

We divided the hormone levels into different subgroups and compared the differences in SRR between subgroups to investigate the differences between SRR under different hormone level groupings. INHB was divided into an abnormal group of 15 pg/mL and a normal group of ≥15 pg/mL according to the range of reference values, with a significant difference in SRR between the groups (*p* < 0.001). AMH was separated into two groups according to the range of reference values: an abnormal group of <2.04 ng/mL and a normal group of ≥2.04 ng/mL. A significant difference was found in SRR between the groups (*p* = 0.02). FSH levels were separated into three groups: normal (11.5 mIU/mL), 11.5–23.0 mIU/mL (2 times the high limit of normal), and >23.0 mIU/mL (2 times above the high limit of normal). The group with the lowest SRR was the one with the highest FSH levels. T was separated into two groups: >5.2 nmol/L and <5.2 nmol/L (the group with the lowest SRR). The SRR between the FSH, INHB, and AMH groups significantly varied (Table 2).

Table 3 shows the binary logistic regression analysis findings of positive sperm retrieval in micro-TESE. The input approach was used to build multivariate models 1 and 2 after modifying the covariates. Because both INHB and AMH were covariate with INHB/AMH, we designed 2 different sets of multivariate analyses. Model 1 excluded INHB/AMH, whereas model 2 excluded INHB and AMH. Model 1 revealed AMH and INHB as independent predictors of being a positive sperm retrieval, INHB (OR1.02, 95% CI: 1.00–1.03, *p* = 0.03), and AMH (OR0.85, 95% CI: 0.75–0.97, *p* = 0.01). Model 2 suggested that INHB/AMH was an independent predictor of positive sperm retrieval (OR1.08, 95% CI: 1.04–1.12, *p* < 0.001).

Figure 2 illustrates the subject workup curves identifying the cutoff values for serum INHB and INHB/AMH in patients with successful micro-TESE sperm retrieval. The cutoff value for INHB was 21.51 pg/mL, with a sensitivity of 49.0%, specificity of 80.3%, Yorden index of 0.29, AUC of 0.64, 95% CI of 0.54–0.73, and *p*-value of 0.005. The best cutoff value for INHB/AMH was 3.19, with sensitivity of 86.3%, specificity of 53.8%, Yorden index of 0.40, AUC of 0.76, 95% CI of 0.69–0.84, and *p*-value of <0.001. The AUC for AMH was 0.37 (<0.5).

INHB is represented by red and INHB/AMH by blue. The area under the ROC curve for the chance diagonal (Gray dotted line, the line segment from 0 to 1) is 0.5.

The usage of INHB and INHB/AMH before performing micro-TESE sperm retrieval in patients with idiopathic NOA improved the net benefit of positive sperm retrieval, according to DCA. A positive net benefit is generated if INHB is >21.51 pg/mL or INHB/AMH is >3.19. DCA showed that INHB/AMH was superior to INHB in terms of sperm retrieval outcomes using INHB and INHB/AMH to identify the net benefit to idiopathic NOA. A 30% likelihood of sperm retrieval was linked with a net benefit of 0 in the absence of any biomarker. The addition of the two biomarkers INHB and INHB/AMH at the 30% threshold improved the net benefit to 0.1–0.2, with a greater net benefit in INHB/AMH scoring than INHB (Figure 3).

The graph contains four different sorts of lines: INHB/AMH (model1) is represented by the blue line, INHB (model 2) is represented by the dashed red line, and the two extreme examples are represented by the solid gray and black lines. The solid gray line on the horizontal one shows that all samples were negative, no one intervened, and there was no net benefit.

## 4. Discussion

The objective of this study was to create a model for men with idiopathic NOA to predict the effectiveness of testicular sperm retrieval using new biomarkers. After doing multivariate logistic regression analysis, we created our prediction model. Finally, predictor—ratio of serum INHB to AMH level—was incorporated into our research model. Serum AMH and INHB have steadily become markers for evaluating the quality of the semen and the function of the male reproductive system in recent years. Earlier studies identical to ours indicated that serum AMH and INHB were related to semen quality [12,13].

Our findings support the earlier conclusion that testicular volume is not a reliable predictor of sperm retrieval outcome in patients undergoing micro-TESE [14,15]. Serum FSH levels are a good predictor of active spermatogenesis. Patients with idiopathic NOA with positive TESE results had considerably lower mean blood FSH levels than those with negative results [16]. Ramasamy et al. revealed that FSH levels did not affect the SRR in micro-TESE, and SRR remained unchanged even when FSH values were significantly elevated [17]. This is consistent with our study results in a binary logistic regression analysis with FSH (*p* = 0.52). Additionally, a meta-analysis revealed that FSH was not a reliable indicator of successful micro-TESE sperm extraction [18], which supports our results. Previous studies reported that AMH levels and AMH/tT ratios achieved independent predictor status for sperm retrieval in micro-TESE with 93% and 95% predictive accuracy, respectively [19]. In contrast to the FSH, LH, E2, and PRL groups, our study found differences in INHB, AMH, or INHB/AMH between the micro-TESE negative and positive groups (Table 1). Reproductive hormones were grouped according to a range of reference values, and the SRRs that differed between groups were INHB, AMH, and FSH (Table 2). Collectively, this points to a connection between reproductive hormones and micro-TESE results. The most likely reason for the differences with the above studies is the difference in the study population. We developed prediction models only for patients with idiopathic NOA, which accounts for the majority of NOA and has widely varying testicular spermatogenesis.

Binary logistic regression analysis was utilized to further investigate the influence of reproductive hormones on micro-TESE, which revealed the predictive effect of INHB, AMH, and INHB/AMH (Table 3). INHB is a heterodimeric glycoprotein that develops in Sertoli cells and controls serum FSH levels in the pituitary gland through a negative feedback mechanism; thus, it may be a more accurate indicator of spermatogenesis than FSH. A Japanese study revealed a significantly higher AUC than that of FSH and testicular volume, making the serum INHB level a useful predictor of testicular sperm presence in males with idiopathic NOA. The cutoff value of serum INHB level was 34.0 pg/mL (sensitivity of 70.6% and specificity of 95.6%) in patients with idiopathic NOA undergoing TESE [20]. AMH is formed in Sertoli cells and mainly serves to degenerate Mullerian ducts and reduce steroid hormone synthesis in males by blocking Leydig cell differentiation. Utilizing AMH or INHB by themselves as a predictor of successful sperm retrieval does not produce the desired outcomes although AMH is an excellent marker for promoting cell development and highly corresponds with sperm concentration and testicular volume. Research on Chinese subjects revealed that INHB had a cutoff value of 28.39 pg/mL (sensitivity of 83.5% and specificity of 79.1%), and its AUC resembled that of FSH [21]. A retrospective Belgian investigation revealed the cutoff concentration for INHB as 13.7 pg/mL (sensitivity: 44.6%; specificity: 63.4%; AUC: 0.51) and combining serum FSH and INHB did not increase the precision of TESE outcome prediction [22]. Our results suggest INHB (AUC = 0.64) as a more accurate micro-TESE sperm recovery predictor than AMH (AUC = 0.37). The intervention inconsistencies may cause disagreement among the outcomes of the above-mentioned research. We discovered that the INHB/AMH had the highest AUC (0.76), sensitivity (86.3%), and specificity (53.8%), indicating that the two are more reliable predictors together (Figure 2).

Our investigation had some limitations. Although our sample size is the largest reported in the literature, the final results may be biased due to the limitations of a single center and limited sample size. The patients received tertiary treatment; thus, the complexity of their conditions may restrict the applicability of our findings. Our prediction model is also not externally validated. Larger research with other centers and demographics is therefore required to support our findings.

## 5. Conclusions

The major outcome of this study was to assess the effects of two preoperative biological markers on the likelihood that males with idiopathic NOA undergoing micro-TESE would retrieve viable sperms. We discovered that serum INHB of >21.51 pg/mL and INHB/AMH of >3.19 in this group more accurately indicated a successful first micro-TESE sperm retrieval. In clinical practice, our findings may provide surgeons with more evaluation strategies to lessen unnecessary surgical trauma in patients with idiopathic NOA.

## Figures and Tables

**Figure 1 jcm-12-00500-f001:**
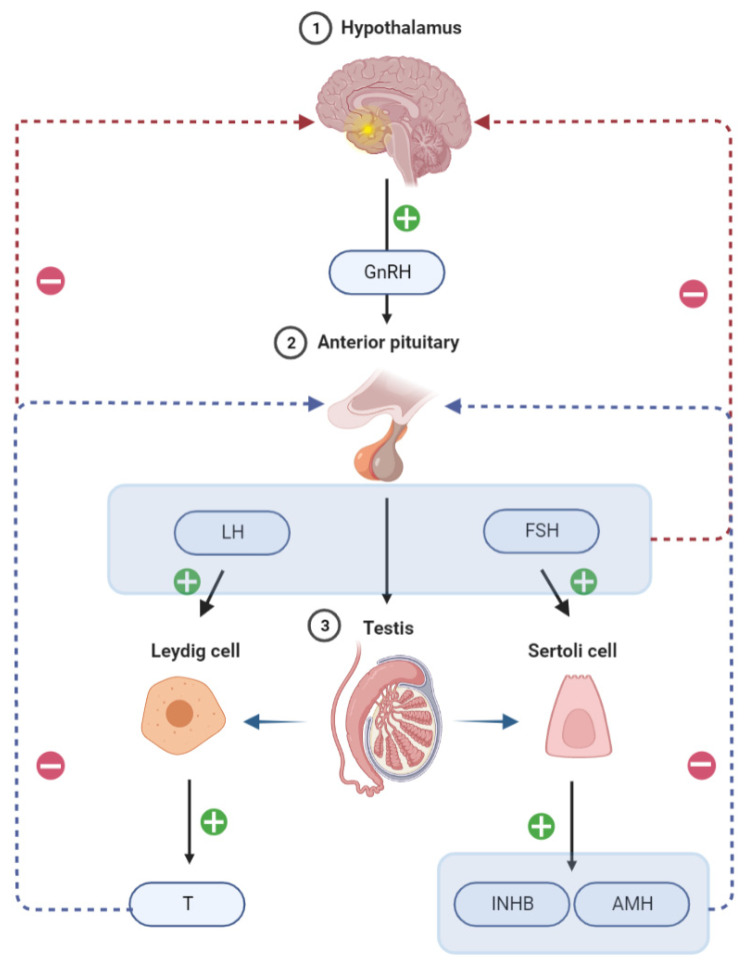
Chematic illustration of the regulation of hormone secretion by the hypothalamic-pituitary-testis axis.

**Figure 2 jcm-12-00500-f002:**
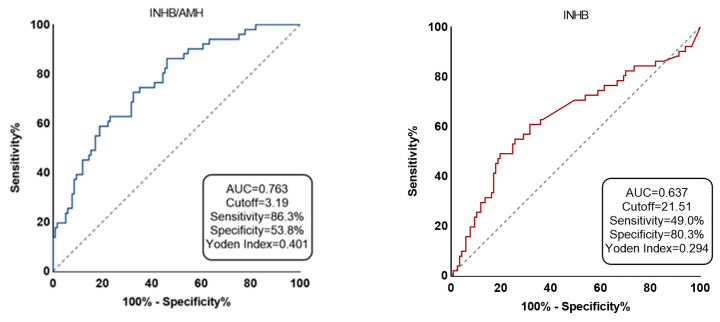
The two predictors’ accuracy is displayed by receiver-operating characteristic (ROC) curve.

**Figure 3 jcm-12-00500-f003:**
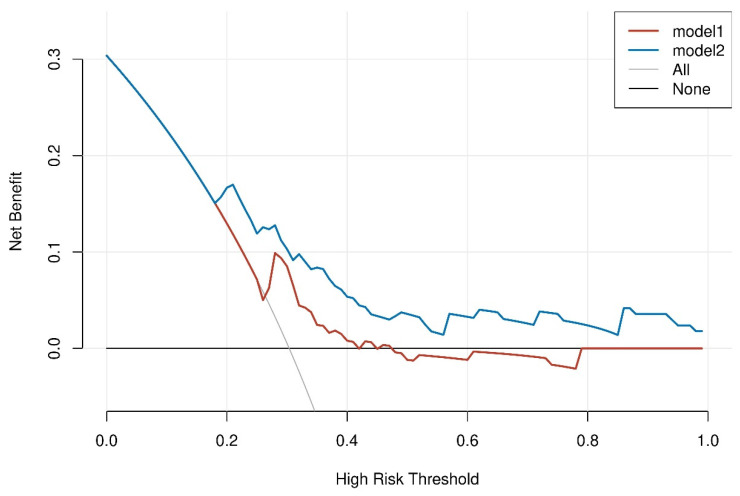
The threshold probability and net benefit are represented by the DCA curve’s horizontal and vertical coordinates, respectively.

**Table 1 jcm-12-00500-t001:** Clinical characteristics and hormonal profile stratified by groups.

Characteristics Median (IQR)	Group1 (*n = 117*)	Group2 (*n = 51*)	*p*
Age (year)	32 (29–34)	32 (30–34)	0.742
Testicular volume (mL)	6 (5–9)	6 (4–11)	0.947
FSH (mIU/mL)	22.60 (16.90–30.70)	16.50 (11.60–25.30)	0.052
LH (mIU/mL)	9.94 (6.87–13.80)	8.11 (5.35–12.00)	0.061
T (nmol/L)	6.86 (4.51–10.30)	6.80 (5.13–9.19)	0.922
Prolactin (ng/mL)	10.73 (7.48–17.24)	11.20 (7.72–18.99)	0.523
E2 (pg/mL)	108.81 (81.88–136.00)	113.00 (83.92–148.56)	0.338
AMH (ng/mL)	3.87 (2.05–7.73)	2.36 (0.65–4.80)	0.007
INHB (pg/mL)	9.98 (5.00–16.27)	17.89 (8.00–36.00)	0.005
INHB/AMH	4.13 (1.63–9.98)	11.50 (4.58–31.55)	<0.001

Group1: Negative sperm retrieval in micro-TESE; Group2: Positive sperm retrieval in micro-TESE; IQR: Interquartile range; FSH: Follicle-stimulating hormone; LH: Luteinizing hormone; T: Testosterone; E2: 17β-estradiol; AMH: Anti-Mullerian hormone; INHB: Inhibin B.

**Table 2 jcm-12-00500-t002:** Comparison of SRR between two groups.

Subgroups	Group1	Group2	SRR	*p*
Inhibin B, *n*				
≤15 pg/mL	85	23	21.3%	
>15 pg/mL	32	28	46.7%	<0.001
AMH, *n*				
≤2.04 ng/mL	29	23	44.2%	
>2.04 ng/mL	88	28	24.1%	0.001
FSH, *n*				
≤11.5 mIU/mL	19	13	40.6%	
11.5–23.0 mIU/mL	40	23	36.5%	
>23.0 mIU/mL	58	15	20.5%	0.001 ^#^
T, *n*				
≤5.2 nmol/L	40	14	25.9%	
>5.2 nmol/L	77	37	32.5%	>0.05
LH, *n*				
≤11.7 mIU/mL	78	38	32.8%	
>11.7 mIU/mL	39	13	25.0%	>0.05

Group1: Negative sperm retrieval in micro-TESE; Group2: Positive sperm retrieval in micro-TESE; SRR: Sperm retrieve rate. FSH: Follicle-stimulating hormone; T: Testosterone; AMH: Anti-Mullerian hormone. ^#^: SRR20.5% vs. other SRR.

**Table 3 jcm-12-00500-t003:** Logistic regression analyses for positive sperm retrieval at micro-TESE.

Factors	Multivariable Model 1			Multivariable Model 2		
	OR	95%CI	*p*	OR	95%CI	*p*
Age	1.014	0.939–1.096	0.718	1.054	0.971–1.144	0.209
AMH	0.850	0.748–0.967	0.013			
INHB	1.015	1.001–1.030	0.031			
PRL	1.024	0.982–1.068	0.268	1.024	0.980–1.070	0.291
FSH	1.014	0.972–1.059	0.515	1.002	0.956–1.049	0.938
LH	0.939	0.849–1.039	0.226	0.914	0.816–1.024	0.123
E2	1.005	0.997–1.013	0.243	1.004	0.996–1.013	0.287
T	0.959	0.875–1.050	0.365	0.949	0.867–1.039	0.260
TV	1.048	0.940–1.170	0.398	1.035	0.928–1.155	0.531
INHB/AMH				1.078	1.040–1.118	0.000

FSH: Follicle-stimulating hormone; LH: Luteinizing hormone; T: Testosterone; E2: 17β-estradiol; AMH: Anti-Mullerian hormone; INHB: Inhibin B; OR: odds ratio; CI: confidence intervals; *p*: statistical significance.

## Data Availability

The datasets used and/or analyzed during the current study are available from the corresponding author on reasonable request.

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
