# Peer review of "Inhibin B-to-Anti-Mullerian Hormone Ratio as Noninvasive Predictors of Positive Sperm Retrieval in Idiopathic Non-Obstructive Azoospermia"

_jcm, 2023, doi:10.3390/jcm12020500_

Round 1
Reviewer 1 Report
General comments to the authors:
From my point of view, the manuscript is well written and the results observed in this study are interesting, however there are some issues that I would like to discuss with the authors:
1. I think I have not seen along the manuscript how the authors calculate/known the INHB/AMH parameter/ratio/value.
2. Comparing the results of Tables 1 and 2, according to Table 1 it appears that group 1 (TESE negative) has higher AMH values on average than group 2 (TESE positive). However, Table 2 establishes new groups where abnormal and lower (< 2.04) AMH values correlate with a better SRR than values within the normal range of the hormone. Could the authors explain why they think they observe these results?
3. I would like to ask the authors if they have pathological anatomy data from the biopsies. And if so, if they have checked the pathological anatomy results of testicular biopsies, in order to see if, for example, hormone levels of patients with a negative micro-TESE result correlate with an increased risk of having a Sertoli cell only syndrome phenotype, and hormone levels that predict success in micro-TESE correlate with an anatomic pathology result of hypo-spermatogenesis.
4. Do the authors believe that in patients with conventional TESE performed, they would observe similar results to those observed in patients who have undergone micro-TESE? Have they verified this?
Specific comments:
Line 42: “The majority of cases of male infertility are caused by azoospermia” Not true, look for prevalence of azoospermia in different reviews. For example: doi: 10.6061/clinics/2013(Sup01)03
Line 44: “due to vas deferens” – better “due to post-testicular ducts”
Line 51: FSH abbreviation is not defined
Line 58-59: “INHB and AMH are secreted specifically by testicular support cells and may be more useful in predicting normal spermatogenesis in the testis than FSH” I would like to see a reference of that sentence.
sperm retrieval rate (SRR) is defined after the first time this abbreviation appear, line 77 instead of line 72.
Line 68: TESE abbreviation is not defined.
Author Response
Response to Reviewer 1 Comments
- I think I have not seen along the manuscript how the authors calculate/known the INHB/AMH parameter/ratio/value.
Response: Thank you very much for these important comments. INHB/AMH is defined as the ratio obtained by dividing the absolute value of INHB by the absolute value of AMH. We explain in the method section of the manuscript, according to your good advice.
- Comparing the results of Tables 1 and 2, according to Table 1 it appears that group 1 (TESE negative) has higher AMH values on average than group 2 (TESE positive). However, Table 2 establishes new groups where abnormal and lower (< 2.04) AMH values correlate with a better SRR than values within the normal range of the hormone. Could the authors explain why they think they observe these results?
Response: Thank you very much for your valuable comments and suggestions on our manuscript. Then, after performing a second accounting and double-checking the data, we discovered that it was accurate and reliable. In fact, SRR is better in people with this hormone below the lower limit of normal values. The fact that AMH was lower in the micro-TESE-positive group than in the negative group is also consistent with our findings in Table 1. Additionally, the individuals in our study sample have idiopathic NOA, which means that their AMH may not be within the normal range. Because of testing procedures and other elements, such as age, reliable reference ranges for serum AMH in men have not yet been established for clinical use. Additionally, our findings imply that AMH lacks predictive power. AMH has an AUC of 0.37 (< 0.5).
- I would like to ask the authors if they have pathological anatomy data from the biopsies. And if so, if they have checked the pathological anatomy results of testicular biopsies, in order to see if, for example, hormone levels of patients with a negative micro-TESE result correlate with an increased risk of having a Sertoli cell only syndrome phenotype, and hormone levels that predict success in micro-TESE correlate with an anatomic pathology result of hypo-spermatogenesis.
Response: Thank you very much for your constructive comments. The value of testicular pathological diagnoses such as Sertoli cell only syndrome in predicting micro-TESE has been reported in the current literature (doi: 10.1111/j.1464-410X.2012.11203.x.). However, because to the added expense and stress of the biopsy, which is entirely the patient's responsibility, very few people have had testicular histopathology. Our study's goal was to see if we could use some non-invasive preoperative indications to try to predict micro-TESE in patients. Thanks again to the reviewers for their comments from a new perspective.
- Do the authors believe that in patients with conventional TESE performed, they would observe similar results to those observed in patients who have undergone micro-TESE? Have they verified this?
Response: Thank you very much for this important comment. The questions from the reviewers were very interesting and worth exploring in depth. Since the current study was retrospective and this group of patients did not choose to undergo conventional TESE first, it is difficult to verify this with our current data. If external conditions permit, we are willing to further explore whether similar results would be obtained with conventional TESE treatment modalities. Thanks.
- Line 42: “The majority of cases of male infertility are caused by azoospermia” Not true, look for prevalence of azoospermia in different reviews. For example: doi: 10.6061/clinics/2013(Sup01)03
Response: Thank you for your valuable reminders. We have rewritten this part according to the Reviewer’s suggestion. Azoospermia, one of the most serious forms of male infertility, is the complete absence of sperm in the ejaculated semen and accounts for about 5% to 20% of male infertility patients. (Please see the revised manuscript at line 42). Thanks.
- Line 44: “due to vas deferens” – better “due to post-testicular ducts”
Response: We are sorry for our incurred writing that we should have avoided. We have made the replacement. (Please see the revised manuscript at line 46). Thanks.
- Line 51: FSH abbreviation is not defined.
Response: We are sorry for bothering you with this kind of mistakes that we should have avoided. Follicle stimulating hormone releasing hormone (FSH). (Please see the revised manuscript at line 53). Thanks.
- Line 58-59: “INHB and AMH are secreted specifically by testicular support cells and may be more useful in predicting normal spermatogenesis in the testis than FSH” I would like to see a reference of that sentence.
Response: Thank you very much for your valuable comments and suggestions. INHB and AMH are not as widely used in clinical practice as FSH, but there is still literature supporting the better predictive value of INHB compared to FSH. Please refer to doi:10.4103/aja.aja_94_18. Our data also suggest an AUC=0.434 for FSH. as shown in the figure below.
- sperm retrieval rate (SRR) is defined after the first time this abbreviation appear, line 77 instead of line 72. Line 68: TESE abbreviation is not defined.
Response: We are sorry for bothering you with this kind of mistakes that we should have avoided. We have corrected it in the manuscript. (Please see the revised manuscript at line 75 and 68). Thanks

Reviewer 2 Report
· The present study has several limitations. The retrospective nature of the study affected the quality of the data. Serum inhibin B and AMH concentrations constitute additional diagnostic parameters in male subfertility as they reflect Sertoli cell function. Stimulated concentrations of serum inhibin B and AMH do not add clinically relevant information in sub-fertile men compared to the basal concentration of these hormones. For the benefit of the reader, the serum inhibin B and AMH concentration could be correlated to testicular histopathology but are not superior to FSH as a predictor of the presence of sperm in testicular sperm extraction (TESE) in men with azoospermia. Reliable reference ranges and thresholds for serum inhibin B in clinical practice have not been established because of the effects of different methodologies and ages. Although a few studies investigated inhibin B concentrations due to the wide reference range and the small number of enrolled subjects, inhibin B use in clinical applications has been limited. Therefore, the existing data are limited and conflicting. Further investigation is required about inhibin B, other fertility-related hormones, semen quality and age in a large scale of infertile and fertile men.
· The manuscript is clear, relevant to the field, presented well-structured, and follows the guidelines of MDPI.
· The cited references are mostly not recent publications (within the last five years). The references are pertinent to the topic. Self-citations do not detect.
· The figures/tables/images/schemes are appropriate and easy to interpret and understand.
· The conclusions are consistent with the evidence and arguments and address the central question.
Author Response
Response to Reviewer 2 Comments
- The present study has several limitations. The retrospective nature of the study affected the quality of the data.
Response: Thank you very much for these insightful remarks. We concur with you entirely. Our study's retrospective design necessitates a low standard of proof. Our research is intended to serve as a foundation for further prospective studies.
- Serum inhibin B and AMH concentrations constitute additional diagnostic parameters in male subfertility as they reflect Sertoli cell function. Stimulated concentrations of serum inhibin B and AMH do not add clinically relevant information in sub-fertile men compared to the basal concentration of these hormones.
Response: Thank you very much for your valuable comments and suggestions. In fact, it is clear from our study's results that in the patients who had successful sperm retrieval, both INHB and AMH were outside the accepted reference range. The following points might be relevant to this. First, because all of the participants in our study had idiopathic NOA, it's possible that their levels of reproductive hormones aren't within normal limits. Second, given their age, testing procedures, etc., the present clinical usage of INHB and AMH in male infertility is not widely used and may lead to a less reliable reference range for men. The standard range for these two hormones is actually quite broad.
- For the benefit of the reader, the serum inhibin B and AMH concentration could be correlated to testicular histopathology but are not superior to FSH as a predictor of the presence of sperm in testicular sperm extraction (TESE) in men with azoospermia.
Response: Thank you so much for your insightful remarks and recommendations. Since both INHB and AMH are specifically secreted by Sertoli cells, theoretically, these two hormones are more responsive to the functional status of Sertoli cells and are likewise related to testicular histopathology. In contrast, FSH is secreted by the pituitary gland and is regulated by negative feedback from subordinate hormones. The predictive value of INHB has also been suggested to be higher than that of FSH in the literature (doi:10.4103/aja.aja_94_18.).
- Reliable reference ranges and thresholds for serum inhibin B in clinical practice have not been established because of the effects of different methodologies and ages. Although a few studies investigated inhibin B concentrations due to the wide reference range and the small number of enrolled subjects, inhibin B use in clinical applications has been limited. Therefore, the existing data are limited and conflicting. Further investigation is required about inhibin B, other fertility-related hormones, semen quality and age in a large scale of infertile and fertile men.
Response: Thank you so much for your insightful remarks and comments. We wholeheartedly concur with the reviews. In clinical practice, there is no trustworthy reference range defined. However, our data suggest that INHB still has the capacity to forecast sperm retrieval outcomes. There is no doubt that larger, even prospective trials are necessary to further validate this.
- The manuscript is clear, relevant to the field, presented well-structured, and follows the guidelines of MDPI. The cited references are mostly not recent publications (within the last five years). The references are pertinent to the topic. Self-citations do not detect. The figures/tables/images/schemes are appropriate and easy to interpret and understand. The conclusions are consistent with the evidence and arguments and address the central question.
Response: Many thanks to the reviewers for their approval of our manuscript.
